# On the Level Sets and Invariance of Neural Tuning Landscapes

**Binxu Wang**          BINXU_WANG@HMS.HARVARD.EDU

**Carlos R. Ponce**          CARLOS@HMS.HARVARD.EDU

*Neurobiology Department, Harvard Medical School, Boston, MA, 02115*

**Editors:** Sophia Sanborn, Christian Shewmake, Simone Azeglio, Arianna Di Bernardo, Nina Miolane

## Abstract

Visual representations can be defined as the activations of neuronal populations in response to images. The activation of a neuron as a function over all image space has been described as a "tuning landscape". As a function over a high-dimensional space, what is the structure of this landscape? In this study, we characterize tuning landscapes through the lens of *level sets* and Morse theory. A recent study measured the *in vivo* two-dimensional tuning maps of neurons in different brain regions. Here, we developed a statistically reliable signature for these maps based on the change of topology in level sets. We found this topological signature changed progressively throughout the cortical hierarchy, with similar trends found for units in convolutional neural networks (CNNs). Further, we analyzed the geometry of level sets on the tuning landscapes of CNN units. We advanced the hypothesis that higher-order units can be locally regarded as isotropic radial basis functions, but not globally. This shows the power of level sets as a conceptual tool to understand neuronal activations over image space.

**Keywords:** level sets, invariance, visual neuroscience, natural image manifold, neural tuning, radial basis function, generative adversarial network, iso-response curve

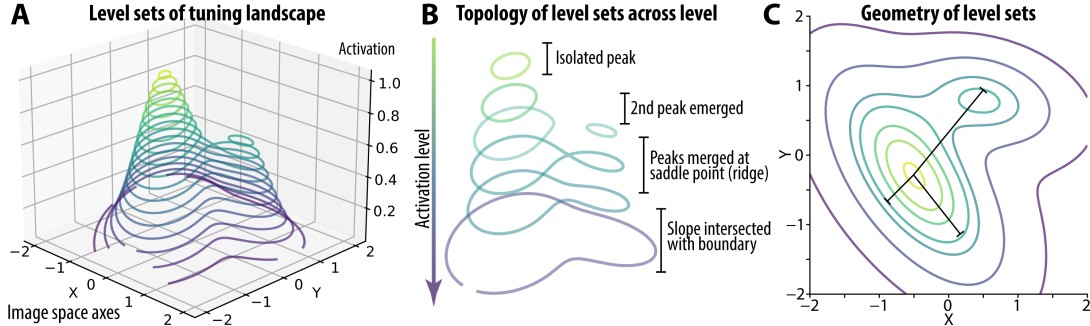

Figure 1: Conceptual schematics. **A.** The level sets of a schematic landscape. **B.** One method for characterizing these sets: tracking topology through levels. **C.** A second method: analyzing the geometry of each level set within and across the connected components.

## 1. Introduction

The visual capabilities of humans and machines depend on the representations of their constituent neuronal units. To a first approximation, a visual representation is a set of feature functions over image space. For primate visual systems, these functions are instantiated

by the firing rates of individual cortical neurons. For automated visual systems such as deep artificial neural networks (ANNs), these functions are instantiated by the activations of hidden units. The feature function is usually a continuous non-negative function of the image: for neurons, the mean firing rate changes continuously with the image; for *in silico* units, activation is a continuous function by design. Thus, we can picture the activation of a neuron or unit over the entire image space as a **tuning landscape** (Wang and Ponce, 2022c). If we do so, one can then ask: what are the image transformations that leave a unit's activations unchanged? In the landscape picture, this translates to finding the iso-response *contours* or **level sets** of the activation function (Figure 1A). Transforming images within each level set will leave activations unchanged. This is a generalized definition of "invariance," the ability of a neuron to respond equally despite nuisance transformations (Ito et al., 1995). At the population level, the intersection of individual neural level sets defines the set of metamers for a population representation (Freeman and Simoncelli, 2011; Feather et al., 2022). Thus, understanding the structure and visual content of level sets can shed light on important neuronal functional properties — a collection of level sets is similar to a computer tomography scan of the tuning landscape. Using tools from Morse Theory (Milnor, 2016), one can track the shape and topology of level sets as a function of the activation, revealing information such as the number of peaks, and the shape of each peak (Figure 1B).

Here, we analyzed the tuning landscape of neurons and units through the lens of level sets. Using data from a novel experimental design (Wang and Ponce, 2022c), we obtained a collection of level sets on 2D tuning maps from neurons in visual cortex. By tracking how the level-set topology changed as a function of activation, we developed a topological signature for each tuning map. This signature was found to change systematically across the visual hierarchy, advancing previous results. We also examined images within the level sets to capture the feature attributes to which the unit was invariant. Finally, using level sets, we studied the high-dimensional tuning landscape of units in an ANN trained for object recognition. We found that higher-order units could be well-approximated by isotropic radial basis functions locally, while globally exhibiting multiple separated peaks. In contrast, the tuning-landscape peaks of mid-level units were much more anisotropic, and better connected by "mountain ridges".

## 2. Formalism and Mathematical Background

Let us consider the tuning function of a neuron or unit, mapping images to non-negative activation values[1] $f : \mathcal{I} \to \mathbb{R}_+$. We assume $f$ is a continuous function of the image. Since image space $\mathcal{I}$ is compact[2], this continuous function has a global maximum and minimum value, according to the extreme value theorem (Rudin et al., 1976). The level set of a tuning function on general image space $\mathcal{I}$ is defined as the set of images evoking the activation level $c$, $\Omega_c = \{I \in \mathcal{I} \mid f(I) = c\}$. In this study, the image space is parametrized by a generative image model $G : \mathbb{R}^d \to \mathcal{I}$ (Dosovitskiy and Brox, 2016), so the level set lies on this $d$

---

1. For neurons, we will consider their mean firing rates in a given time window and disregard dynamics, *e.g.,* fluctuation and adaptation effects
2. When regarding an image as a red-green-blue (RGB) pixel array, the space of images is bounded.

dimensional image manifold.

$$\Omega_c = \{I = G(z), z \in \mathbb{R}^d \mid f(G(z)) = c\} \tag{1}$$

When $c$ is a *regular value* of the function $f$, the level set on a $d$-dim manifold is a $(d-1)$-dim hypersurface (per the implicit function theorem Hatcher (2000)). When $c$ is a *critical value* (*e.g.,* the value of a peak or a saddle point), the level set may contain discrete points. From Morse theory (Audin et al., 2014), if an interval $[a, b]$ contains only regular values, then the level set $\Omega_a$ and $\Omega_b$ has the same topology (it is *homeomorphic*). Using this fact, when we find that the topology of a level set changes in an interval $[a, b]$, we can conclude there is a critical value (*e.g.,* a second tuning peak) lying in this interval (for examples, see Figure 1B). Thus, tracking the level set as a function of activation level can characterize the landscape.

For a high-dimensional image space, fully representing and characterizing a $d - 1$ dimensional hypersurface is intractable. So, we used two strategies to study these: in Sec.3, we consider a tuning function on a 2D sub-manifold in the image space; then, the level sets become 1D curves—easier to study. In Sec.4, we sample discrete points from the high-dimensional level sets and estimate their properties (Figure 1C).

## 3. Level Sets of Neuronal Tuning *in vivo*

In this section, we study the tuning landscapes of visual cortex neurons using level sets. Because most datasets comprise arbitrary, irregularly sampled natural images, they do not facilitate neuronal response interpolation needed to calculate a level set. In contrast, generative image models allow for the dense sampling of visually similar stimuli and perfectly suited level set extraction — thus, we used these generative models for our experiments.

**Experimental Design.** The activity of visual neurons in V1, V4, and posterior inferior temporal (pIT) cortex were recorded using chronic microelectrode arrays (Microprobes for Life Sciences). A generative image model $G$ with 4096 latent dimensions (Dosovitskiy and Brox, 2016) parameterized a naturalistic image manifold. We used this manifold to characterize neuronal tuning functions and to find each site's tuning peaks. In each experimental session, one cortical site was selected as an experimental target (which could be a single neuron or multi-unit cluster), and then the neuron-guided image synthesis or *Evolution* approach (Ponce et al., 2019) was used to search for images that maximized firing rate (Figure 2A). After responses converged (Loshchilov, 2014; Wang and Ponce, 2022a), the evolutionary trajectory in the latent space was analyzed by principal component analysis (PCA). Then, a 2D hemisphere was created in the subspace spanned by the top three PC vectors of the trajectory (Figure 2B). We denoted the coordinate map of hemisphere as $\varphi : [-\pi/2, \pi/2] \times [-\pi/2, \pi/2] \to \mathbb{R}^{4096}, (\theta, \phi) \mapsto z$. Latent codes were sampled regularly on the longitude-latitude $(\theta, \phi)$ grid, and then mapped to images by $G$. This 2D hemisphere was designed to section through the peak and to visualize the shape of tuning around optimal stimuli[3]. Finally, the sampled images were shown to the subject and neuronal responses $r$ recorded. Each session led to a 2D tuning map, which could be understood as the com-

posite function $f \circ G \circ \varphi : (\theta, \phi) \mapsto z \mapsto I \mapsto r$, mapping spherical coordinates $(\theta, \phi)$ to neuronal activations $r$ (firing rate averaged over $[50, 200]$ms after stimulus onset).

**Level Set Extraction.** As shown in Wang and Ponce (2022c), neurons from V1 to pIT showed smooth tuning maps on the image manifold (Figure 2C). Spherical spline interpolation was used to obtain a smooth function over the hemisphere (see appendix B.1 for details). We sampled $K = 21$ levels uniformly between the *min* and *max* values of the tuning map and extracted these level sets (Figure 2D,E).

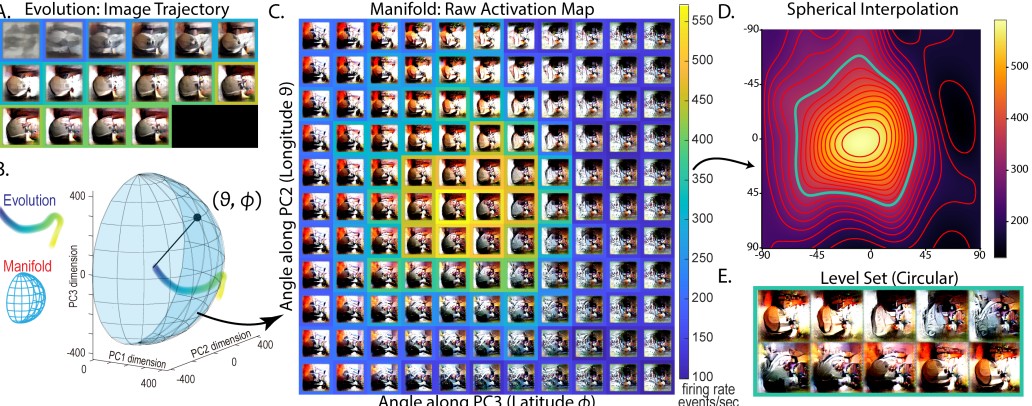

Figure 2: Example *Manifold* experiment (*in vivo*) (Wang and Ponce, 2022c). **A.** Process of image optimization as driven by a V4 multiunit in monkey B (*Evolution* experiment). Image frame color denotes mean firing rate evoked per generation (same color scale as **C**). **B.** Latent space geometry of *Evolution* trajectory and *Manifold* sampling grid, plotted in the 3D subspace spanned by the PC1,2,3 vectors of the *Evolution* trajectory, adapted from Wang and Ponce (2022c, Figure 2). **C.** Raw neuronal response as a tuning map over the *Manifold* images. **D.** Spherical interpolation of tuning map and all level sets. **E.** One circular level set with corresponding image loop.

### 3.1. Topological Signature of Level Sets on Tuning Maps

First, we examined the topology of each level set. Since each level set $\Omega_c$ is a 1D manifold, it has to be the union of segments that are topologically homeomorphic to either a circle $S^1$ or a line $(0, 1)$ (Hatcher, 2000). Based on this fact, for each level set, we computed the number of segments (connected components) that are homeomorphic to circles $N_S$ and to lines $N_L$. We tracked these numbers $N_S, N_L$ as a function of the activation level $c$. We call this the *topological signature* of the level sets.

We found that generally, at a high activation levels, the topology of the level set was a circle $S^1$. At lower activations, the level set became more elongated and fractured. We tracked

---

3. This specific way of sampling was informed by the spherical geometry of the latent space of the generator $G$ (Wang and Ponce, 2022a, Figure B.1). In this latent space, angular distance was a better correlate of perceptual distance than linear distance; changing the vector norm largely changes image contrast. So, we fixed the vector norm and sample vectors on a sphere.

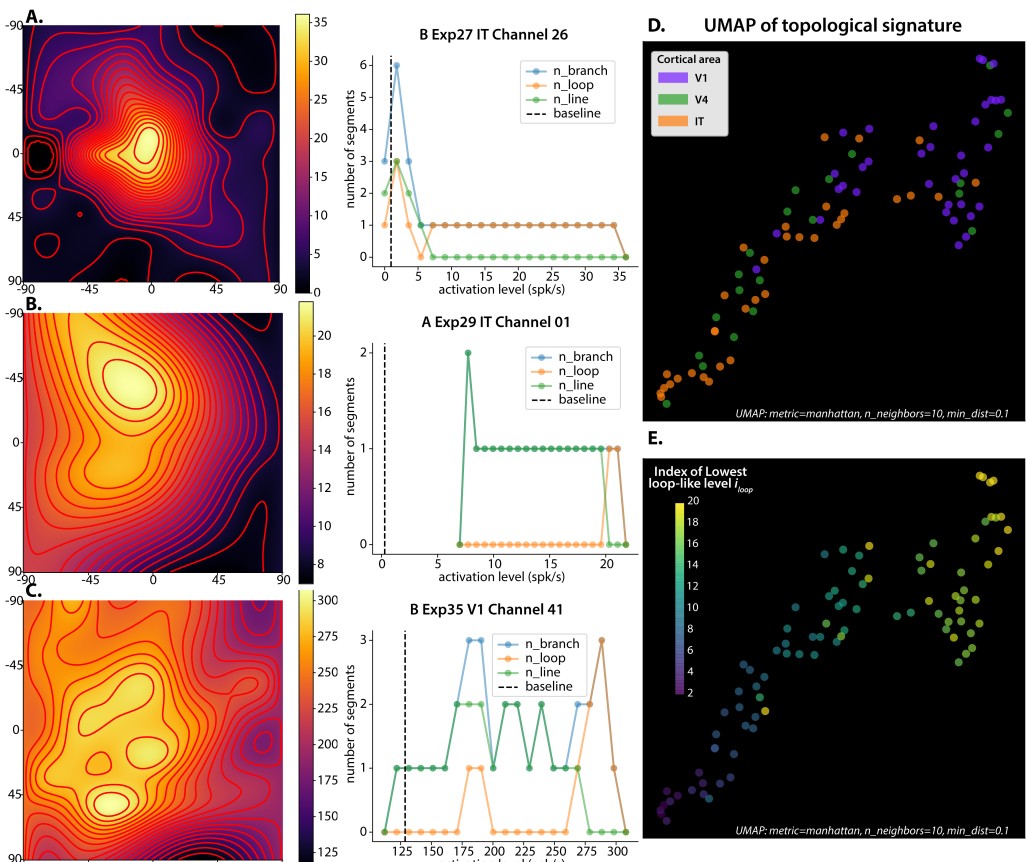

Figure 3: Topological signatures of level sets *in vivo*. **A-C.** Example tuning maps and their topological signatures. (Left panel) $K$ level set contours on the tuning manifold. (Right panel) Solid lines and dots denote the number of branches, loops and lines on each level as a function of neuronal activation level. **A.** A sharp single peak surrounded by many circle segments. **B.** A broader single peak, with more line segments. **C.** An irregular, multi-peak landscape, with multiple circular and line segments. **D.** UMAP plot of individual cortical sites (color indicates area of origin). **E.** Same plot with color coding of index $i_{loop}$.

the level set topology down from the peak and defined an index $i_{loop}$ as the *lowest level* where the level set was still a single circle. This index quantifies the level where the level set changed its topology, either by intersecting the boundary or going through a second peak. Since all levels between $i_{loop}$ and $K$ are homeomorphic to a loop, we can infer that above the level $i_{loop}$, the landscape looks exactly like a bell (homeomorphic to a disk $\mathbb{D}^2$).

To see how this topological signature translated to the landscape's geometry, we investigated a few examples in Figure 3. A sharp isolated peak on the tuning map ($A$) was reflected by many level sets with a single circular component at a high activation level. A broader peak ($B$) was encoded by multiple line-like level sets, since the mountain slope intersected with the boundary of the hemisphere. A more fractured and multi-peak landscape ($C$) led to multiple connected components, even at high activation levels.

We then performed a population analysis using dimensionality reduction (UMAP) (McInnes et al., 2018) on the topological signatures of all tuning maps ($n = 90$). We used the number

of loops $N_S$ and lines $N_L$ at $K$ levels as the feature and Manhattan distance as the metric. The UMAP analysis revealed a spectrum of topological features across all neurons (Figure 3D). We found that the index $i_{loop}$ strongly correlated with this spectrum (Spearman correlations with UMAP coordinate 1,2 were 0.923, $p = 3 \times 10^{-38}$ and 0.82, $p = 8 \times 10^{-23}$), providing one interpretation of this map. When we labeled the UMAP points by their visual area, we found that the progression of visual hierarchy (V1 to V4 to pIT) mapped onto this spectrum (Figure 3E). The index $i_{loop}$ decreased across the ventral hierarchy (Spearman correlation of $i_{loop}$ with hierarchical level was $-0.565$, $p = 6.5 \times 10^{-9}$; one-way ANOVA of $i_{loop}$ and cortical area, $F = 23.01$, $p = 9.1 \times 10^{-9}$ ). Simply put, more IT neurons have tuning maps as in example $A$, while more V1 neurons have maps as in example $C$. In topological terms, a narrow isolated tuning peak resulted in more loop-like level sets and lower $i_{loop}$, while multi-peaks on a highland led to non-connected level sets at a high level, leading to a higher $i_{loop}$.

As a comparison, we performed the same Manifold experiments on units from a deep neural network (for details see B.4). The same level set analysis revealed a trend of topological signature change across the layers, similar to that across the ventral hierarchy (for *in silico* experiments, Spearman correlation of $i_{loop}$ with the layer depth was $-0.589$, $p = 0, df = 9502$. For qualitative comparison of the mean topological signature, see Figures 8 and 7).

## 3.2. Emergent invariance in level sets

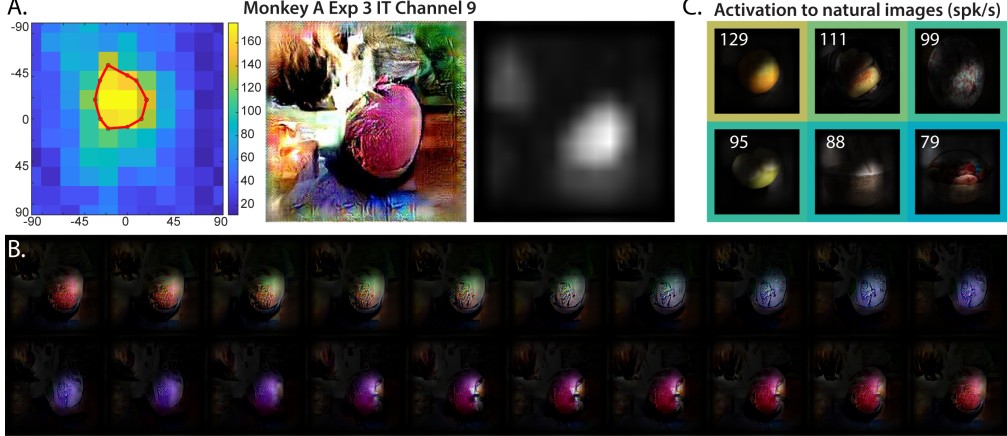

Figure 4: Example of an interpretable image transformation of one level set *in vivo*. A circular level set that mapped to a color/hue circle in the image space. **A.** (**Left**) The circular level set on the tuning map of an IT neuron in monkey A. (**Middle**) The peak image on the manifold. (**Right**) The feature attribution mask highlighting the image region that correlated the most with the neural activation. For details see Sec.B.2. **B.** The image content of the level set, with the feature attribution mask emphasizing the region important to the neuron. **C.** Natural images that were also strongly activating. Firing rates are labeled at the corner and encoded in the color frame.

Next, we examined the visual content of each level set to study its associated feature invariance. We found that the image transformations of a subset of level sets were highly

interpretable. For example, in one case, a pIT neuron guided the *Evolution* search towards a red-pink round object (Figure 4A). We extracted and visualized the level set at a high level (80% max). Intriguingly, the circular level set looping around the peak corresponded nearly perfectly to a *hue circle* with approximately the same spherical shape as the "object". Thus we could describe the image transform along this level set as "changing the color of the object by rotating along the hue circle". More formally, we can define a group action of a $\mathbb{S}^1$ circle group transforming the image by traveling on the circle. We also examined the same neuron's responses to natural images in the same session (Figure 4C). Though their activation level was not as high as the peak image, they had different colors: an orange, a green apple, brown toast, etc. This suggests that this IT neuron was relatively invariant to the object's color. More generally, since circular level sets were common on the landscape, especially around tuning peaks, we conclude that it is relatively simple to create image transformations to which a neuron will be invariant. Crucially, we also noted that many level sets corresponded to image transformations that were hard to summarize in simple words (see Figure 9). We will discuss the implications of this observation in the end.

## 4. Level Sets of Deep Neural Network Units

Next, we characterized the tuning landscape of deep neural network units on the same generative image manifold. Without *in vivo* experimental constraints, we were able to explore the full image manifold and characterize level sets more comprehensively. This could generate testable hypotheses for neural representations *in vivo*.

**Methods.** Here we used an optimization method to sample from level sets and characterize them using these samples. Since each level set $\Omega_c$ can have multiple connected components, we were interested in both local and global properties (i.e. structure of one component or across components). We analyzed *local* properties by first finding a local maximum, and then searching for level set images starting from that maximum. *Global* properties were analyzed by searching for level set images starting from random initializations. Pictorially, a local search ends at points near the connected component around a given peak; a global search ends with points lying on different "peaks" (Figure 5A). By analyzing the distance structure among images within a component (*local*) or across components (*global*), we could characterize the geometry of the tuning landscape.

More concretely, we first performed an initial *Evolution* to obtain one activation maximizing image $I^* = G(z^*)$ and its code $z^*$ as the base. Then, we searched for level set images using the objective Eq.2. The first term penalized the deviation of activation from the level $c$; the second term minimized or maximized the perceptual distance from a target $I^*$, depending on the sign of $\beta$. For a local characterization, the search was initialized from the peak $z_*$; for a global characterization, from a random vector. We used the perceptual image distance LPIPS (Zhang et al., 2018) as $D$, and masked the images by their receptive field masks (methods detailed in appendix B.3). For each unit, we performed the search under three main conditions: local search, minimizing image distance ($\beta = 5$); local search, maximizing image distance ($\beta = -5$) ; global search, maximizing image distance ($\beta = -5$). We also performed the local and global search with $\beta = 0$ as baseline conditions.

$$\arg \min_z |f(G(z)) - c| + \beta D(G(z), I^*) \tag{2}$$

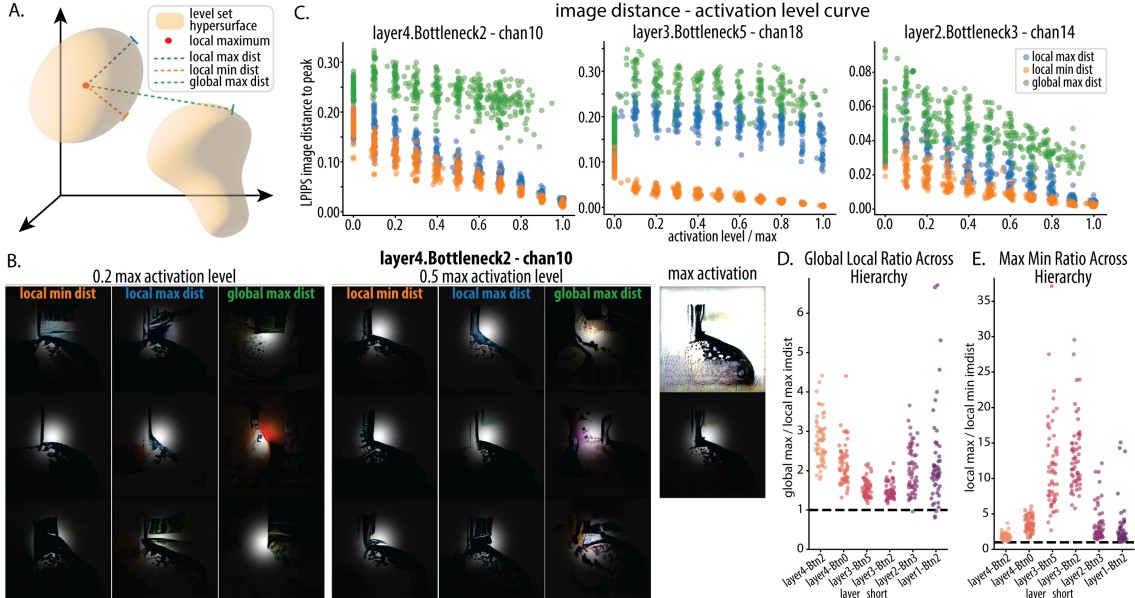

Figure 5: Characterizing level sets for *in silico* units. **A.** Conceptual schematics showing level sets as hyper-surfaces and our strategy for characterizing them. **B.** Image samples from two level sets $(0.2, 0.5)$ for a unit in layer 4 (images masked by its receptive field). Each column shows three samples obtained under the same search condition (*local min, local max, global max image distance*). The base image (local maximum) is shown in the rightmost column. **C.** Examples of image distance - activation level curve, in three different layers. **D.** Ratio of global versus local max image distance in a level set, as a function of layer. **E.** Ratio of local max versus min image distance in a level set, as a function of layer.

**Model network**  We chose the adversarially trained ResNet50-robust (He et al., 2016; Engstrom et al., 2019), as the *in silico* visual hierarchy to be compared with the ventral stream. This choice was made based on its high ranking on the Brain-Score leaderboard (Schrimpf et al., 2018) at the time of this study. We sampled six layers and 60 units per layer; for each unit, we searched for 11 activation levels, with 5 different search conditions, and 40 repetitions, totaling 792,000 optimization runs.

### 4.1. Global Topology of Level Sets

First, we focused on the basic topological property of path-connectedness of the level set. We made the assumption that if a level set was path-connected, a gradient-based search within the set could explore every point in it. So, we compared the level-set samples found by local versus global searches, both maximizing distances to the base image $I^*$. Specifically, we compared their image distances to the base image. The ratio of image distance between global and local search estimates the ratio of the distance between connected components and the "radius" of one component. Indeed, we found that the global search resulted in a larger diversity of level-set images, which were further apart from the initial peak $I^*$ (Figure 5B,C). Throughout the hierarchy, the ratio of the mean image distance between the global and the local search was always greater than one (Figure 5D). Thus, we conclude that

these level sets were generally *not path-connected*, and the tuning landscapes exhibited peaks at multiple locations which are not connected by a ridge. We also note that the global/local ratio for hierarchically higher units was larger, and the ratio for mid-hierarchical units was closer to one (`layer3.B2`, mean±std $1.53 \pm 0.24$, `layer3.B5`, $1.62 + 0.30, N = 60$). This suggests that the landscape of mid-level units was "more connected" — meaning that when traveling along one level set component, the farthest distance one can reach was close to the farthest one can get globally. Pictorially, this means that *the peaks of mid-level units were connected* by "mountain ridges". In contrast, *the peaks of high-level units were more isolated*, and each local component was farther apart.

### 4.2. Local Geometry of the Level Sets and Tuning Peak Isotropy

Next, we examined the local geometry of the level sets. We computed the ratio between the *max* and *min* image distances $D$ to the peak from one local component $\bar{\Omega}_c$. $\lambda_{maxmin}(c) = \frac{\max_{I \in \bar{\Omega}_c} D(I, I^*)}{\min_{I \in \bar{\Omega}_c} D(I, I^*)}$ The max-min ratio $\lambda_{maxmin}$ is closely related to the condition number of the Hessian matrix at the tuning peak, given a local quadratic approximation to the neural tuning function. The numerator points to the flattest direction to the level set, while the denominator points to the steepest direction from the peak. So, using this index we can quantify the perceptual anisotropy of the tuning peak.

We found that for mid-layer units, the ratio was large, *i.e.,* there was a large gap between the minimum and maximum image changes that induced a certain level of activation decrease (Figure 5CE). In contrast, for the higher order visual representations (especially the penultimate layer in ResNet50-robust), the ratio was surprisingly close to 1 (mean±std, $1.77 \pm 0.46, N = 60$) — so even the "fastest" descent and "slowest" descent path from the peak had a similar slope. This highlights the isotropic nature of the peak.

In summary, we can interpret these results in the context of the radial basis function (Poggio and Girosi, 1990). It has been proposed that in a multidimensional space, the tuning of neurons could be viewed locally as a radial basis function (Wang and Ponce, 2022c). Here, we validated that the tuning landscapes of higher-order visual units can be locally approximated well by isotropic radial basis functions: any direction deviating from the peak has a similar slope. However, this approximation does not hold globally, as images sampled from other components of the level set did not fall on the same trend Figure 5. This suggests we should not limit global characterization of tuning to just one radial basis function, and certainly not to readily interpretable tuning variables (Wang and Ponce, 2022c). In contrast, for mid-level units, there exist some very flat and also very steep paths deviating from the peak, illustrated by an elongated level set with large aspect ratio. Thus even locally, their tuning landscapes are anisotropic and exhibit certain "untuned" directions. This result is consistent with the model proposed in Wang and Ponce (2022c, Figure 6), where higher visual neurons have more tuned axes than the middle ones and appear to be more isotropic.

## 5. Discussion

In machine learning, the concept of level sets has been a classic tool for representing and analyzing shapes (Osher et al., 2004). In neuroscience, it has been used under the name of

"iso-response method" to understand the computation that integrates two or three features in *early* sensory neurons (e.g. locust auditory receptor, retina ganglion cells, V1 neurons, Gollisch et al. (2002); Rust et al. (2005); Horwitz and Hass (2012); Gollisch and Herz (2012)). In this work, we extended this tradition and applied this concept to analyze the geometry of tuning landscapes of neurons throughout the visual hierarchy. Thanks to the advance of image generative models and closed-loop experiment techniques, we can obtain level sets both for higher visual neurons and in the high-dimensional space of complex natural images. As we showed, the geometry of level sets exhibited trends across the visual hierarchy that were consistent *in vivo* and *in silico* (Figures 8 and 7). This adds another quantitative tool for comparing representations in brains and machines, in addition to representation similarity analyses (Kriegeskorte and Wei, 2021).

The next stage for visual neuroscience is to define how neurons respond across naturalistic image transformations, specifically characterizing their selectivity and invariance. When considering neuronal responses across a large image manifold (via the *tuning landscape* perspective), it appears trivial to define a group action (i.e. image transformation) of a $\mathbb{S}_1$ circle group on a local image domain, to which the neuron activation is "invariant." The group action is to transform the image by rotating on the circular level set. More generally, the level set of a function on a $d$ dimensional manifold is a $d-1$ dim hypersurface. We hypothesize that if a neuron is tuned to $d$ dimensions, one should find a $d-1$ dimensional hypersphere-like level set around its local maximum, where the neuron will be invariant to any transformation on the hypersphere. This may have implications for the development of equivariant neural networks.

There are compelling examples where the level sets on the tuning landscape contained interpretable image transformations (*e.g.* color of the object), but more often than not, there are no easy labels for the transformations characterized by the set. This is likely because neurons/units just learn useful features, and signal distances to these features, regardless of interpretability. By being tuned to multiple features, the diminishing presence of one feature can be compensated by the persistence of another. Thus, nameable image transformations (*e.g.,* rotation) may be a small fraction of all the image transforms to which a unit or neuron is invariant.

## Acknowledgments

We thank Mary Carter for help in monkey care and data collection, Kaining Zhang for helpful discussion on analysis and algebraic topology, and Matthew Farrell for his talk that inspired this work.

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

## Appendix A. Related Work

**Iso-response Set of Tuning Functions**   There is a rich history of analyzing the level sets of tuning functions for biological neurons and artificial units. Much of this literature can be found under the terms of *iso-response sets*. For *in silico* units, Paiton et al. (2020) analyzed the iso-response surfaces of units in a sparse coding network, and found that one direction of adversarial perturbations was orthogonal to the surface. This makes sense since this was the "steepest" tuning direction. For biological neurons, Rust et al. (2005, Figure 5) analyzed the iso-response contours of primate V1 neurons on the 2D plane spanned by the activities of two linear filters (subunits). By inspecting the shape of the ellipsoidal contours and their alignment with the axes, the authors inferred the form of nonlinear operations combining the linear filters. Horwitz and Hass (2012) used a closed-loop experiment to find the iso-response surface for V1 neurons in the 3d color space spanned by cone-photoreceptor activity space. By analyzing the geometry of these 2D surfaces (*e.g.*, some being planar, others ellipsoidal or cup-shaped), they identified different linear or nonlinear computations used by these V1 neurons used to integrate cone input signals. Gollisch and Herz (2012) reviewed experimental results on the iso-response curve: the locust auditory receptors, with respect to the intensity of two pure tones in superposition (Gollisch et al., 2002); retina ganglion cells, with respect to the contrast levels of two patches; and V1 neurons with respect to colors (Horwitz and Hass, 2012). They proposed that by analyzing the geometry of the "iso-response" curve on the plane spanned by two stimulus features, one could disambiguate different feature integration mechanisms. They also suggested that closed-loop optimization was crucial to estimate these level sets. These previous works focused on early sensory neurons or receptors and analyzed contours in simple stimulus spaces. Here, we extended this approach to neurons in higher visual cortices (V4 and pIT) within complex naturalistic image manifolds. This advance was made possible by our closed-loop experimental approach that could find neurons' "preferred" stimuli and by the use of an image generator that could manipulate images smoothly.

Shifting to studies of perception, Zetzsche et al. (1999) found the set of stimuli that had a *just noticeable distance* (JND) from a given Gabor patch on a 2D image space. By comparing the shape of the contour formed by these stimuli under different parametrizations (Zetzsche et al., 1999, Fig.5,6), the authors found that certain parametrization schemes (such as polar coordinates) were more fitting to perception because they led to more isotropic, circular contours. This argument can also apply to the more isotropic tuning peaks for higher visual units in our study. It suggests the latent space parametrization of images is more "natural" for higher visual neurons like IT.

**Level Sets in Machine Learning**    A level set is a classic concept in multivariate calculus and geometry. In machine learning, it is a popular tool for representing $d - 1$ dimensional hyper-surface data with a scalar function on $d$ dimensional spaces (Osher et al., 2004). The level set method is also a classic approach for image segmentation, which represents the boundary of an object by the level sets of a function on the 2D plane. More recently, the neural implicit function method represents 2D shapes and surfaces by the level sets of a 3D volumetric function parametrized by a neural network (Chen and Zhang, 2019). In many cases, the function learned by the network is the *signed distance function* (SDF) to the object surface (Park et al., 2019). In these works, the level set is a tool to learn and represent a target signal. In contrast, in our work, we used level sets to analyze representations already learned in biological and computational visual systems.

**Object Manifold in Neural Representation**    One similar line of work is the one studying the geometry of object manifolds (Chung et al. (2018); Cohen et al. (2020)). Our conceptual pictures complement each other: they consider the manifold in the neural activation space, traced by the neural activations to the different views of the same object. We considered the level set manifolds in the image space, traced by the different images that evoked comparable neuronal responses.

## Appendix B. Method details

### B.1. Details for Spherical Interpolation

In Wang and Ponce (2022c), images and associated responses were organized on a regular grid of spherical coordinates $\theta, \phi$. Thus, the spherical geometry needed to be taken into account when interpolating the responses. We used the `RectSphereBivariateSpline` method in `SciPy` to interpolate and smooth neuronal response on the hemisphere. We realized that neuronal response noise would affect all downstream analyses, so we set the smoothing parameter `s` as the sum of the squared standard error of the mean (sem) of response over the tuning map. This balanced the reconstruction error and the smoothness of the map.

For contour extraction, we used the marching squares algorithm (Lorensen and Cline, 1987) implemented in `skimage.measure.find_contours` function for each contour line. Finally, we re-parametrized each contour line with arc length parametrization and sampled the curve uniformly. Namely, the latent codes were sampled with equal angular distance along the curve. The resulting level sets were shown in Figure 2D,E.

### B.2. Feature Attribution Mask

For the *in vivo* level set analysis, we masked the image with a "feature attribution mask", which can be estimated using a recently developed method Wang and Ponce (2022b). Conceptually, this is a data-driven way to find the image region and pattern that correlate the most with the neuronal responses. Briefly, this method took in all the image and response pairs in the Evolution trajectory and correlated every unit activation in a convolutional neural network with the neuronal response. Then the covariance tensor was factorized into the combination of a few rank-1 factors, each one being the product of a spatial mask and a feature vector. Finally, we fit a simple linear model to determine the relative weights of

the factors. We used these weights to combine the spatial masks and obtained the alpha mask on the image as in Figures 4 and 9.

### B.3. Receptive Field Estimation

For the *in silico* level set analysis, the image was masked by a "receptive field mask". Simply, we estimated the receptive field by computing the gradient from the activation to the image pixels $\nabla_I f(I)$, then we repeat this by sampling white noise images 200 times and averaging these gradient maps

$$\mathbb{E}_{I \sim unif[0,1]} \nabla_I f(I)$$

This average gradient map is normalized and smoothed to be the receptive field mask, as in Figures 5, 10, 11, 12 and 13

### B.4. Manifold Experiment *in silico*

As in Wang and Ponce (2022c), we conducted Evolution and Manifold experiments on units sampled along the ResNet50-robust. We sampled units from the center of feature maps of every channel in the CNN, totaling 38012 units from 10 layers (`relu`, `layer1.B1`, `layer2.B0`, `layer2.B2`, `layer3.B0`, `layer3.B2`, `layer3.B4`, `layer4.B0`, `layer4.B2`, `fc`, B is the abbreviation for `Bottleneck` layer). In each experiment, the receptive field of the unit was measured; the image was resized to fit the receptive field of the unit. Then the same *Evolution* and *Manifold* experiments were applied to the unit to obtain its activation maximizing image and the 2D tuning maps around the image. Then the tuning maps were analyzed by the same level set topology analysis as in Sec. 3.

## Appendix C. Extended Results

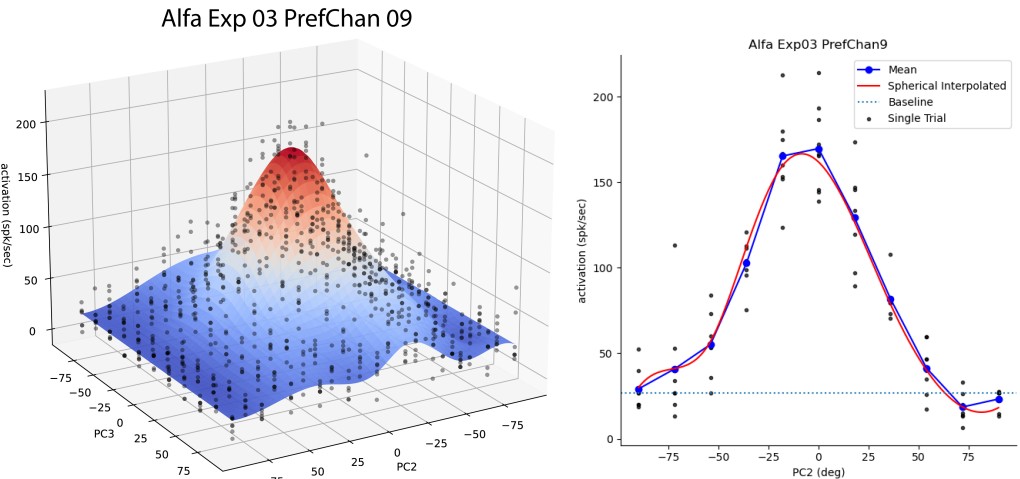

Figure 6: Neuronal fluctuation *in vivo*. **Left**. The surface plot shows the spherical interpolation of a tuning map *in vivo* (a neuron in IT of monkey A). 3d scatter shows the single trial neuronal responses to each image. **Right**. 1d section through the 2D tuning map, showing the single trial firing rates (scatter), mean response (blue), and the spherical interpolated activation curve (red). This motivates us to use the smoother interpolated curve when computing the level sets.

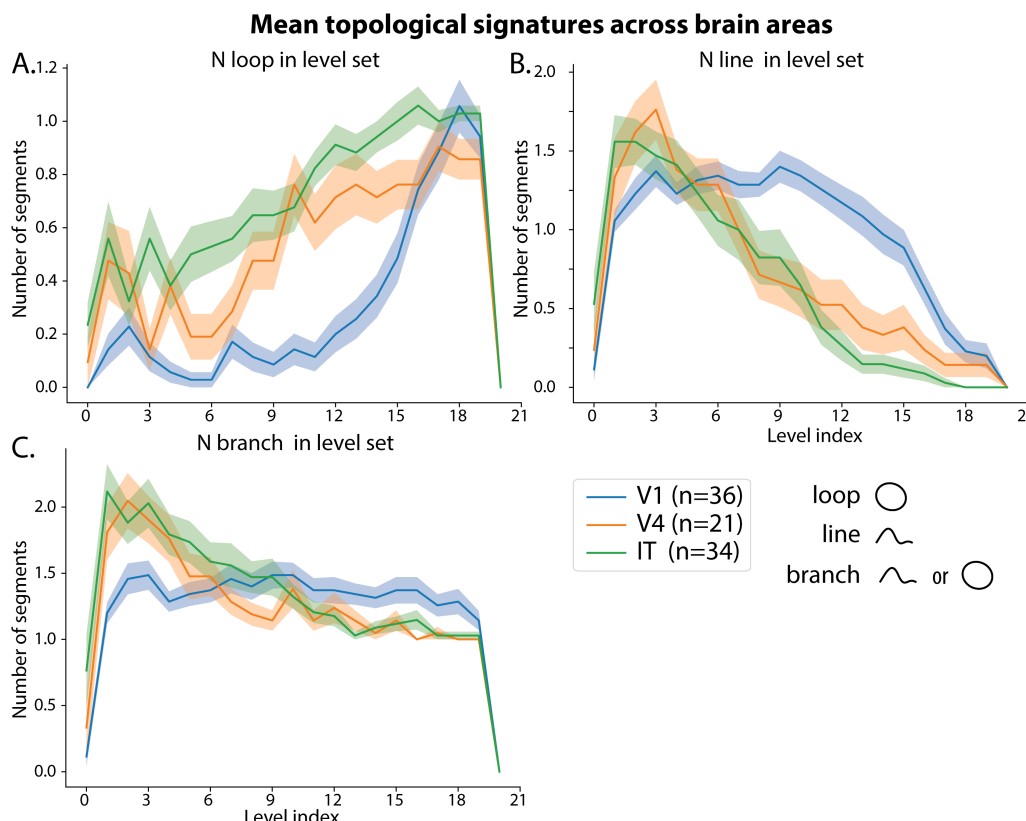

Figure 7: Mean topological signature per visual area *in vivo*. Each curve shows the mean±sem of number of loops (**A**), lines (**B**), and branches i.e. connected components (**C**) as a function of level index; each color corresponds to a given visual area (V1, V4, or IT).

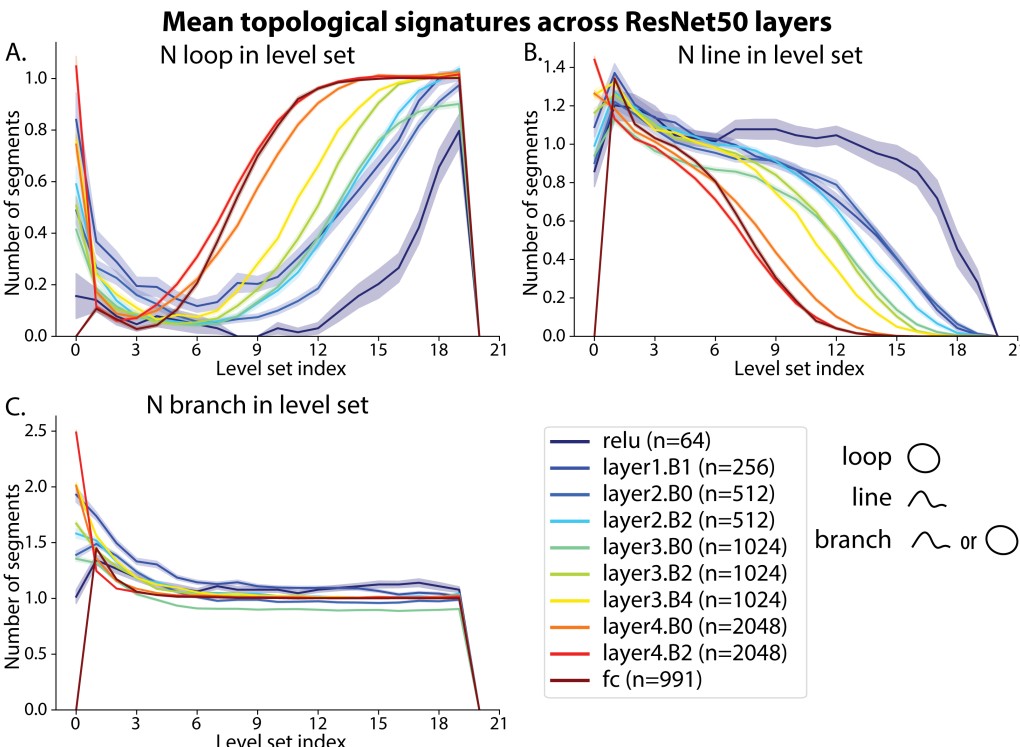

Figure 8: Mean topological signature per layer in network ResNet50-robust. Each curve shows the mean±sem number of loops (**A**), lines (**B**), and branches i.e. connected components (**C**) as a function of level index; each color corresponds to a given layer.

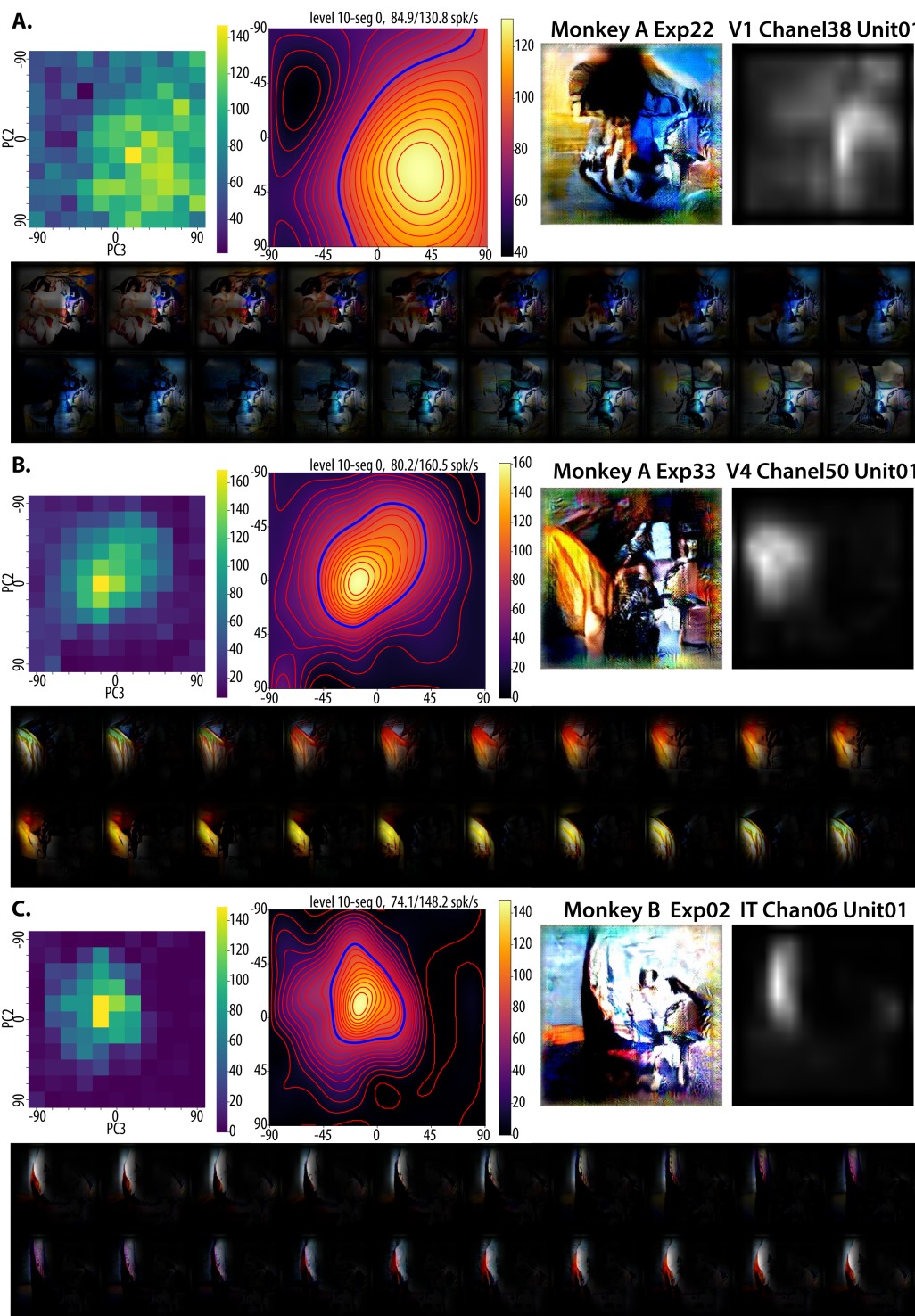

Figure 9: Examples of level sets lacking readily interpretable image transformations. Examples from **A.** V1, **B.** V4, and **C.** IT are showed in the three panels. Figure layout in each panel is similar to Figure 4.

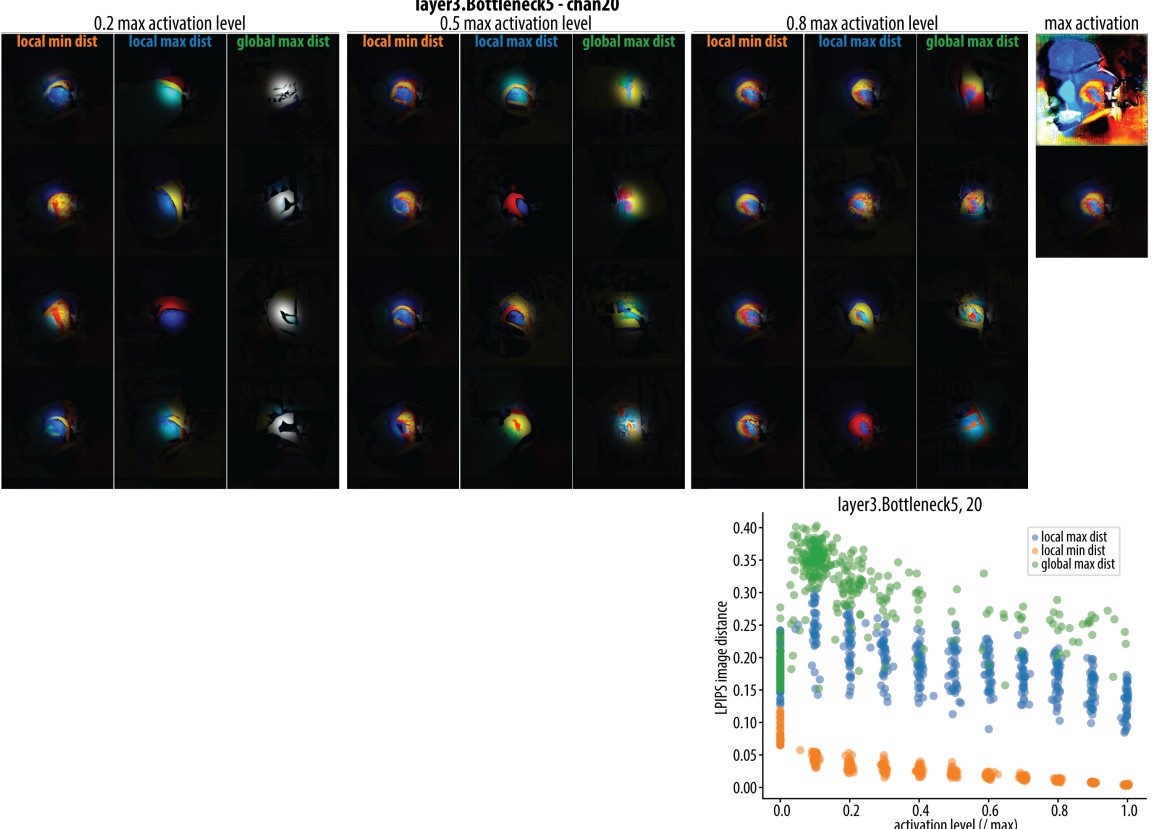

Figure 10: *in silico* Level set image samples for an mid-level unit in layer 3. **Upper** Level set images for three levels are shown (0.2, 0.5, 0.8 max activation level). Samples obtained with the same objective are shown in a column. Three objectives were *local min*, *local max* and *global max* image distance search. **Lower** Summary of all level set images for this unit. The large gap between the local min and local max distance highlights the anisotropy of the peak. The smaller gap between the local max and global max signifies the different connected components of the level set are not too far away. For examples with another organization, see Figure 11, Figure 12, Figure 13.

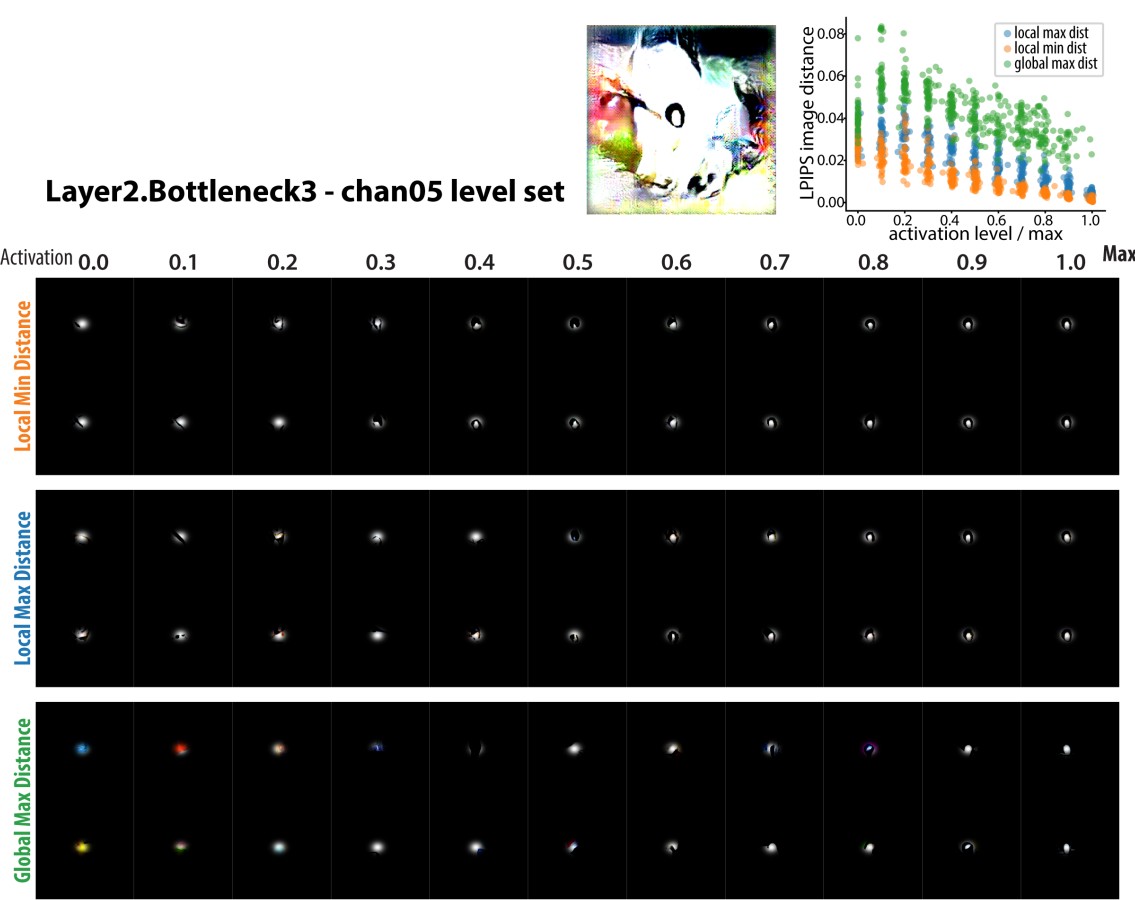

Figure 11: *in silico* Level set image samples across activation levels of a unit in Layer2-B3. **Upper left** Initial peak, base image $I^*$. **Upper right** Summary of all level set images for this unit. **Lower**, Each block collect the level set images obtained in one optimization condition: from top to bottom, *local min distance, local max distance, global max distance*. Column from left to right shows level set images for 0.0 to 1.0 max activation level, two images per level per condition.

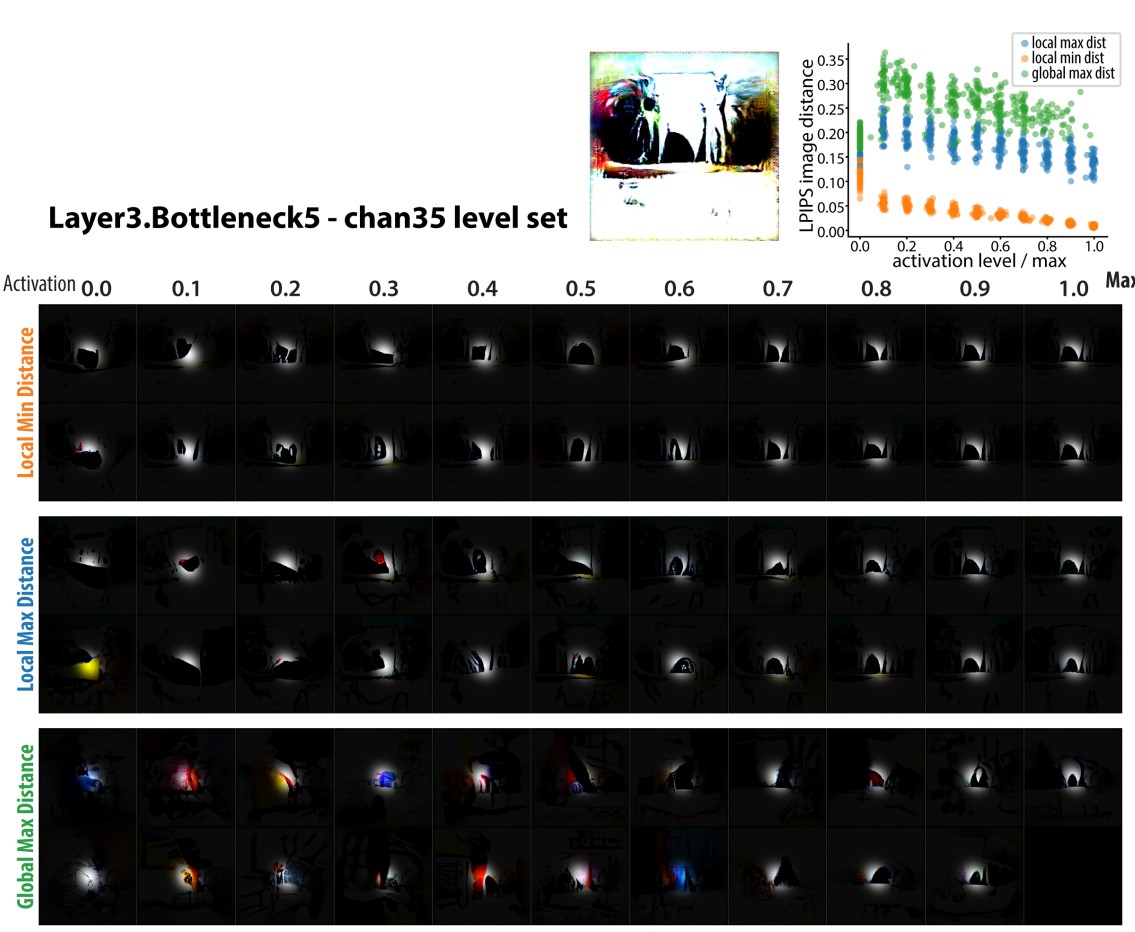

Figure 12: *in silico* level-set image samples across activation levels of a unit in Layer3-B5. Same organization as in previous figure.

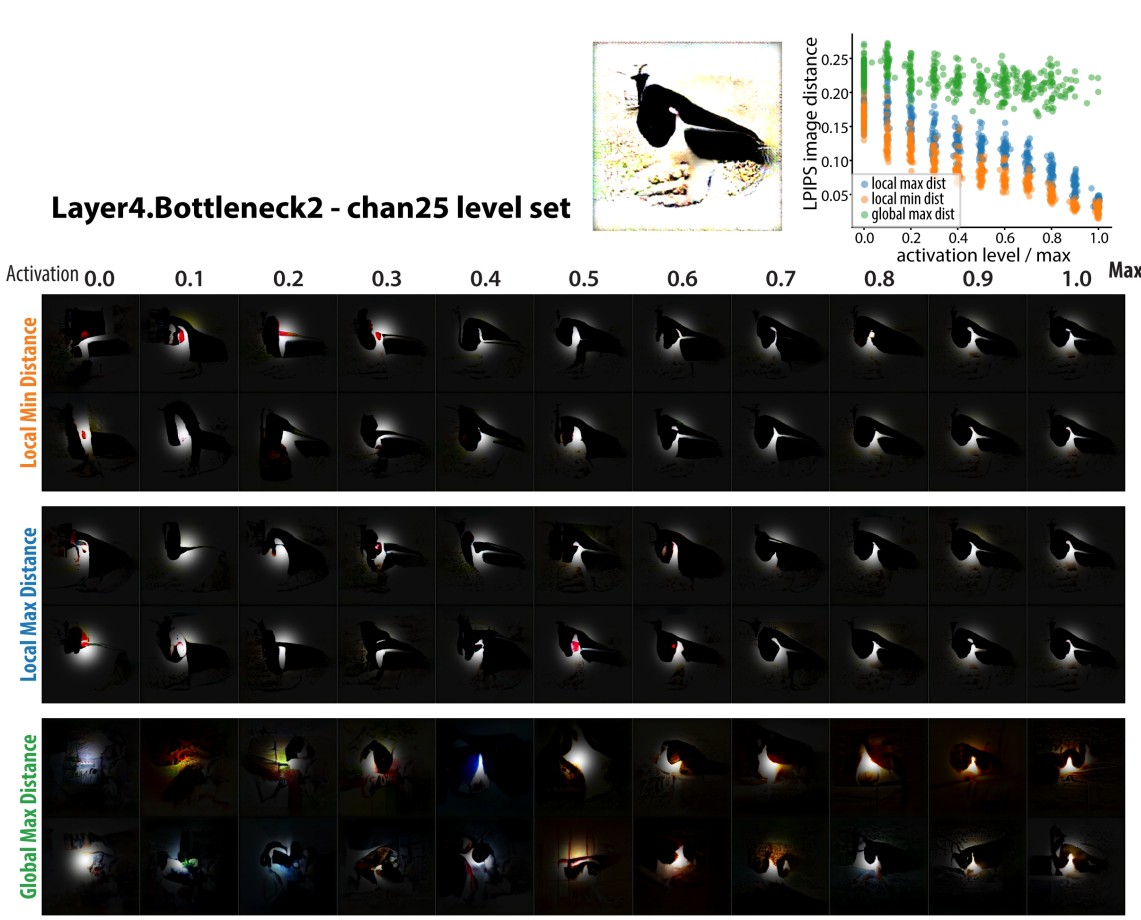

Figure 13: *in silico* level-set image samples across activation levels of a unit in Layer4-B2. Same organization as in previous figure.

