# OpenReview forum: "On the Level Sets and Invariance of Neural Tuning Landscapes"
_NeurIPS.cc/2022/Workshop/NeurReps — NeurReps 2022 Poster_

### Official Review · Reviewer_C2X9 · 2022-10-10
**Interesting analysis of neural responses surfaces with limited topological applications**

**Confidence:** 4
**Soundness:** 3
**Presentation:** 3
**Contribution:** 3
**Overall Rating:** 6

**Summary:**

The paper proposes studying neural response surfaces and performing topological analysis of the level sets at different response heights.

**Questions:**

What is meant with ‚robust‘ in the abstract?

The first sentence of the abstract suggests that neural activity is causal for perception, and that neural networks imitate this process. These are two highly controversial claims and I believe that the paper does not need these questionable statements to be introduced.

What are the different lines in Fig 3, the legend is confusing?

Shouldn‘t the effect of i_loop over ventral areas rather be assessed with an ANOVA?

Fig 4: what is the black and white third image in subplot A?

Is the interpretations (hue circle) of that one unit a fluke? How many are really interpretable, if this is mere chance (there is no explicit disentanglement in the objective), then this should not be presented as a perk of the method.

Is it correct to say higher level visual Neurons are more invariant, i.e., more peaked tuning curves?

**Limitations:**

What about changing tuning in real biological Neurons (e.g. neural drift)?

What exactly is the benefit that the topological perspective offers here? For instance, the connectedness in section 4.1 doesn‘t seem to require recursion to topological terms to understand. What is the specific insight gained with this approach here? I.e., what have learned about artificial and biological networks with this approach?

**Recommended Decision:**

3: Accept

**Relevance:**

3: Solid fit

**Strengths And Weaknesses:**

The figures are all visually very clear and pleasing.

The reliance of this work on previous papers by Wang and Ponce makes it a little unclear how novel this is when compared to those previous results.

More prior work should be cited on iso-response contours, but also on closed-loop experimental studies (e.g., https://www.nature.com/articles/s41593-019-0517-x)

The construction of the hemisphere for sampling images was not very clear and also lacks motivation, please improve exposition and motivation.

Figures 6 and 7 in the appendix need better legends and more explanation

In section 4, methods, please consider citing prior work on diverse image search (http://openaccess.thecvf.com/content_ECCV_2018/papers/Santiago_Cadena_Diverse_feature_visualizations_ECCV_2018_paper.pdf)

**Submission Track:**

Proceedings Paper (9 Page)

---

> ### Author Response · Authors · 2022-10-31
> **Thanks for the constructive comment and questions**
>
> Thanks for the suggestions and constructive questions of the reviewer!
> We answer each question point by point.
>
> **Strengths And Weaknesses:**
> > The construction of the hemisphere for sampling images was not very clear and also lacks motivation, please improve exposition and motivation.
>
> We appreciate the insight. In the revision, we added pertinent explanations in the main text. Briefly, we used a generative image model which has a spherical geometry in its latent space (Wang & Ponce 2022, GECCO). This means, firstly, that the norm of the latent vector  controls the image contrast, and so travelling along the radial direction changes image contrast, while travelling along the angular direction on the sphere changes the spatial configuration of the image pattern. We showed that the angular distance in the space correlated more with the perceptual distance of the generated image measured by LPIPS (Zhang et al. 2018). Because of this geometric fact, we found that sampling equidistantly (in terms of Euclidean distance on a plane) would not generate perceptually equidistant image sequences. In contrast, sampling equidistantly in terms of angular distance on a sphere did generate perceptually uniform image sequences (Wang & Ponce 2022, GECCO).Thus, when designing the *in vivo* experiments, we chose to sample on a sphere of fixed norm to keep the image contrast at similar level. Moreover, we sampled the longitude-latitude grid to make the image samples more uniformly distributed.
>
> > Figures 6 and 7 in the appendix need better legends and more explanation
>
> We have now re-written the legends to clarify them. We have also added more graphical information to illustrate the geometric motifs behind each plot.
>
>
> **Questions:**
> > What are the different lines in Fig 3, the legend is confusing?
>
> We added this explanation to the legend:
>
> Different lines in the right panel denote the number of branches (connected component of any kind), lines (connected components that are homeomorphic to line segment) and loops (components that are homeomorphic to a circle S1) in each level set. These integer numbers are plotted as a function of activation level for each neuronal sit in A,B,C.
>
>
> > Shouldn‘t the effect of $i_{loop}$ over ventral areas rather be assessed with an ANOVA?
>
> Indeed, we could apply ANOVA to $i_{loop}$, and we added this test in the text.
> However, in the absence of targeted contrasts, the ANOVA only shows that the number is different across ventral areas. We also wanted to show that this number was progressively larger for higher visual areas. To test that hypothesis, we used a Spearman correlation to show the relationship between the estimated number and hierarchical level.
>
> > Fig 4: what is the black and white third image in subplot A?
>
> The third image highlights image regions that correlate most with the Evolution (or activation maximization process), as determined using features in a pre-trained neural network and a matrix factorization method. The goal is to segment the synthetic image to show the most important regions to the neuron.
>
> This is detailed in the Sec. B.2 in appendix, but as a brief overview, essentially we pass the sequence of images created during the Evolution to a pretrained neural network. Then we find the units in the pretrained neural network that correlate the most with the neuronal firing rate response. Finally, we factorize the correlated tensor with non-negative matrix factorization to highlight the regions most associated with the spike rate.
>
> The method has not been published in a full article, but it is also described in our short paper at the 2022 Cognitive Computational Neuroscience conference:
>
> @article{Wang2022FactorizedCM,
>  title={Factorized convolution models for interpreting neuron-guided images synthesis},
> author={Binxu Wang and Carlos R. Ponce},
> journal={2022 Conference on Cognitive Computational Neuroscience},
> year={2022},
> doi={10.32470/ccn.2022.1034-0}
> }
>
> **Limitation**
> > What about changing tuning in real biological Neurons (e.g. neural drift)?
>
> A valid concern. This phenomenon is of great interest to us. For the present study, each set of images forming a Manifold were collected in one day. Responses were stable in that day and although there was some response adaptation happening due to the repetition of images (i.e., neural responses to each image decay slightly over time), this adaptation did not qualitatively change the geometry of the map. Interestingly, for other studies, we have collected synthetic images from the same channel in the chronically implanted array for over a year, and we find that for the same given channel, the images continue to encode the same basic motifs, especially when compared to other channels. So for now, our preliminary data suggests that the neural drift is minimal in these regions of the brain (V1, V4 and posterior IT).

---

### Official Review · Reviewer_ocgk · 2022-10-14

**Confidence:** 5
**Soundness:** 3
**Presentation:** 4
**Contribution:** 3
**Overall Rating:** 7

**Summary:**

The paper explores the utility of measuring and visualizing level sets of biological and artificial neuron response recordings. They analyze data recorded in a previous study from monkey neurons as well as activations from an artificial neural network. The results suggest that the response landscape geometry varies between functionally distinct neural populations, such as V1 and V4, and between artificial network layers.

**Questions:**

**Questions:**

(1) You say in the methods that “We assume f is a continuous function of the image. Since image space I is compact, this continuous function exhibits a local and global optimum, per the extreme value theorem”. How is it that you are able to guarantee a single global optimum? For example, an “energy” complex cell model (f(x) = sqrt(ax^2 + bx^2), where a and b are a quadrature pair) would exhibit maximum activation for a continuous set of inputs. Does the possible existence of many global optima influence your analysis?

(2) Your method for finding the level sets assumes a particular target activation value (variable ‘c’ in eq 1) as input. Given that the (mean) biological and artificial neuronal activations are considered continuous, how did you choose the target values below the maximum? Assuming you choose some discrete and uniform set of activations between [0, maximum], how does changing the discretization influence the topology and geometry analysis? What sensitivity does the meta (UMAP & Fig 5 D/E) analysis have to the number of discretization levels?

(3) The authors state in the discussion that “the geometry of level sets exhibits trends across the visual hierarchy that were consistent in vivo and in silico.” Which results support the consistency between the two experiments?

(4) As a control experiment to understand the impact of the interpolation, it would be beneficial to report the mean & std of the activations of the ANN units for the level set images?

**References:**
[1] Gollisch, T., & Herz, A. V. (2012). The iso-response method: Measuring neuronal stimulus integration with closed-loop experiments. Frontiers in Neural Circuits, 6, 104.

[2] Horwitz, G. D., & Hass, C. A. (2012). Nonlinear analysis of macaque v1 color tuning reveals cardinal directions for cortical color processing. Nature Neuroscience, 15(6), 913.

[3] Rust, N. C., Schwartz, O., Movshon, J. A., & Simoncelli, E. P. (2005). Spatiotemporal elements of
macaque v1 receptive fields. Neuron, 46(6), 945–956.

[4] Zetzsche, C., Krieger, G., & Wegmann, B. (1999). The atoms of vision: Cartesian or polar? Journal of the Optical Society of America A, 16(7), 1554–1565.

[5] Paiton, D.M., Frye, C.G., Lundquist, S.Y., Bowen, J.D., Zarcone, R. and Olshausen, B.A., 2020. Selectivity and robustness of sparse coding networks. Journal of vision, 20(12), pp.10-10.

[6] Cadena, Santiago A., et al. "Diverse feature visualizations reveal invariances in early layers of deep neural networks." Proceedings of the European Conference on Computer Vision (ECCV). 2018.

**Limitations:**

An explicit overview of the limitations of the work is not provided, and would be beneficial in the Discussion.

**Recommended Decision:**

2: Borderline

**Relevance:**

4: Highly relevant

**Strengths And Weaknesses:**

**Originality:**
There has been a considerable amount of work measuring the level sets of biological & artificial neurons. For example, see [1-3] for studies on biological neurons and [4, 5] for studies on artificial neurons. Citations of and within those works provide a rich history of studying neurons from the context of level sets. You might also be interested in [6], which explores “invariant images” in artificial neural networks that may or may not lie along a connected contour.

That being said, the submitted work is unique from anything I have read on the subject. Specifically, I do not believe others have used “topology tracking” or the local/global min/max distance methods for characterizing the level sets. Additionally, I believe the insight on the change in “connectedness” between peaks (via the isotropic radial basis function analysis) across the cortical hierarchy is original.

**Quality:**
I found the paper to be technically sound and the methods appropriate (although see clarification questions below).

**Clarity:**
The paper is well written and the methods are communicated well enough given the restricted space. However, I would like to see more of an effort to place this study in the context of prior work (see Originality, above).

**Significance:**
While the idea of using level sets to analyze neural data is not new, the submitted work offers a unique analysis approach and utilizes animal data from a prior study’s reportedly novel experimental design. Thus, the presented work is significant in that it provides an additional method for analyzing and comparing artificial and biological neurons that will hopefully continue to inspire the field to explore level sets and response geometry.

**Submission Track:**

Proceedings Paper (9 Page)

---

> ### Author Response · Authors · 2022-10-30
> **Thanks for the constructive feedbacks**
>
> We are grateful for the careful, detailed and supportive comments of the reviewer.
> Thanks for pointing out the previous literature on the level set analysis of the biological and artificial neurons. We will add them into our camera ready version and integrate ourselves into this tradition of tuning level set analysis.
>
> Here are our answers to the specific questions
>
> > (1) You say in the methods that “We assume f is a continuous function of the image. Since image space I is compact, this continuous function exhibits a local and global optimum, per the extreme value theorem”. How is it that you are able to guarantee a single global optimum? For example, an “energy” complex cell model (f(x) = sqrt(ax^2 + bx^2), where a and b are a quadrature pair) would exhibit maximum activation for a continuous set of inputs. Does the possible existence of many global optima influence your analysis?
>
> Q1:
> The reviewer is definitely correct, we didn’t write it clearly. Arbitrary continuous function can exhibit any number of local or global maxima on a compact space.
> The reviewer is correct that this could have been written more clearly. Arbitrary continuous functions can exhibit any number of local or global maxima on a compact space. But when designing the study, we accounted for this and assumed that there might be multiple “peaks” on the image manifold. In fact, one specific reason to use topological analysis (i.e., level set tracking) was to detect the hypothetical existence of multiple peaks,  through level set topology changes. In the geometry analysis, we also hypothesized that multiple peaks could exist. By comparing local minimum- vs. global minimum distances, we were prepared to detect multiple peaks on the tuning landscape of hidden units on the image manifold.
>
> > (2) Your method for finding the level sets assumes a particular target activation value (variable ‘c’ in eq 1) as input. Given that the (mean) biological and artificial neuronal activations are considered continuous, how did you choose the target values below the maximum? Assuming you choose some discrete and uniform set of activations between [0, maximum], how does changing the discretization influence the topology and geometry analysis? What sensitivity does the meta (UMAP & Fig 5 D/E) analysis have to the number of discretization levels?
>
> Q2
> The reviewer is correct that this choice of “slicing level” will affect the result. In the worst case scenario, when there is a local maximum lying between two slicing levels, then this peak will be invisible to our topological analysis. To account for this, we used uniform spacing levels between the min and max activations of the observed tuning map. We used 21 levels (including the min and max).  But we will show the stability of the UMAP result is stable relative to the choice of level number.
>
> > (3) The authors state in the discussion that “the geometry of level sets exhibits trends across the visual hierarchy that were consistent in vivo and in silico.” Which results support the consistency between the two experiments?
>
> Q3
> Thank you for pointing this out. The consistency refers to the increasingly “isolated” peaks for higher visual areas comparing to lower visual areas. This isolation is showed by more loop like level sets $i_{loops}$ at higher activation levels for higher visual cortices.
> This conclusion was supported by the Appendix, by the topological signatures plotted in Figure 6 (ventral stream areas) and Figure 7 (ResNet50). The in silico experimental design was detailed in B.3. In response, we will work to strengthen the link between the statement and the associated supporting evidence.
>
> > (4) As a control experiment to understand the impact of the interpolation, it would be beneficial to report the mean & std of the activations of the ANN units for the level set images?
>
> This is a great suggestion. To show the necessity of an appropriate interpolation scheme, we visualized the mean and standard deviation of biological neuronal activations to individual images in the supplementary section(Fig. X). For the ANN, there was no interpolation confound, since it was designed to be noise-free.

---

### Official Review · Reviewer_4NGN · 2022-10-15
**Level sets are level sets**

**Confidence:** 2
**Soundness:** 2
**Presentation:** 2
**Contribution:** 2
**Overall Rating:** 5

**Summary:**

Level sets are used to characterise the "tuning landscape", i.e., the mean activation as a function of system input, of _in vivo_ neurons and ANNs for low-level visual tasks. Heuristic features of the measured data are used to justify the claim that level sets are a meaningful way to analyse neural activation data.

**Questions:**

-

**Limitations:**

-

**Recommended Decision:**

2: Borderline

**Relevance:**

3: Solid fit

**Strengths And Weaknesses:**

A well-defined composition of generative model and neuron activation creates an empirically meaningful target for topological data analysis. Most interestingly, a correlation is found between topological signature and cortical hierarchical level. Beyond that, I find it difficult to assess how much contribution was achieved either in experimental procedure or process insight, and I think that the results would be difficult to reproduce for a lack of statistical and algorithmic information.

**Submission Track:**

Proceedings Paper (9 Page)

---

> ### Author Response · Authors · 2022-10-30
> **Thanks for comments. Reproducing code & statistical procedure attached**
>
> Thanks for commenting and summarizing our work.
> We will argue that reproducing our results and experiment is fairly simple and the experiments could click and run.
>
> The official code base for conducting the *in silico* experiments and analyzing the *in vivo* data is here.
> https://github.com/Animadversio/Tuning-Manifold-Level-Sets
>
> The in vivo neural data were deposited in Open Science Framework.
> https://osf.io/gpzm5/

---

### Decision · Program_Chairs · 2022-10-21

Accept (Poster)